# ADAM17 is an essential attachment factor for classical swine fever virus

Fei Yuan[1☯], Dandan Li[2☯], Changyao Li[3], Yanan Zhang[2], Hao Song[4], Suhua Li[2], Hongkui Deng[5]*, George F. Gao[1]*, Aihua Zheng[2,6,7]*

1 CAS Key Laboratory of Pathogenic Microbiology and Immunology, Institute of Microbiology, Chinese Academy of Sciences, Beijing, China, 2 State Key Laboratory of Integrated Management of Pest Insects and Rodents, Institute of Zoology, Chinese Academy of Sciences, Beijing, China, 3 College of Veterinary Medicine, China Agricultural University, Beijing, China, 4 Research Network of Immunity and Health (RNIH), Beijing Institutes of Life Science, Chinese Academy of Sciences, Beijing, China, 5 Peking University Stem Cell Research Center, Department of Cell Biology, School of Basic Medical Sciences, Peking University Health Science Center, Beijing, China, 6 Key Laboratory of Tropical Translational Medicine of Ministry of Education, School of Tropical Medicine and Laboratory Medicine, Hainan Medical University, Haikou, China, 7 College of Life Science, Henan Normal University, Xinxiang, China

☯ These authors contributed equally to this work.
* hongkui_deng@pku.edu.cn (HD); gaof@im.ac.cn (GFG); zhengaihua@ioz.ac.cn (AZ)

**Data Availability Statement:** All relevant data are within the manuscript and its Supporting Information files.

**Funding:** A.Z. received all the grants. This project was funded by the National Key Plan for Scientific

## Abstract

Classical swine fever virus (CSFV) is an important pathogen in the swine industry. Virion attachment is mediated by envelope proteins E$^{rns}$ and E2, and E2 is indispensable. Using a pull-down assay with soluble E2 as the bait, we demonstrated that ADAM17, a disintegrin and metalloproteinase 17, is essential for CSFV entry. Loss of *ADAM17* in a permissive cell line eliminated E2 binding and viral entry, but compensation with pig *ADAM17* cDNA completely rescued these phenotypes. Similarly, *ADAM17* silencing in primary porcine fibroblasts significantly impaired virus infection. In addition, human and mouse *ADAM17*, which is highly homologous to pig *ADAM17*, also mediated CSFV entry. The metalloproteinase domain of ADAM17 bound directly to E2 protein in a zinc-dependent manner. A surface exposed region within this domain was mapped and shown to be critical for CSFV entry. These findings clearly demonstrate that ADAM17 serves as an essential attachment factor for CSFV.

## Author summary

Classical swine fever virus (CSFV) is a highly pathogenic RNA virus belonging to *Flaviviridae* family that can cause deadly classical swine fever (CSF) among pigs. In this study, we identified ADAM17 as a binding partner for CSFV envelope protein E2 using biochemical approaches. Knockout of ADAM17 rendered permissive porcine cells resistant to CSFV infection, which could be reversed by complementing ADAM17 cDNA. The metalloproteinase domain of ADAM17 directly bound to CSFV E2 in a zinc-dependent manner *in vitro*, within which a 45 amino-acid region played key roles in mediating CSFV entry. Discovery the essential attachment factor for CSFV will provide a mechanistic

Research and Development of China
(2016YFD0500303), the National Science and
Technology Major Project (2018ZX10101004) and
National Natural Science Foundation of China,
General Program (81871687). The funders had no
role in study design, data collection and analysis,
decision to publish, or preparation of the
manuscript.

**Competing interests:** I have read the journal's
policy and the authors of this manuscript have the
following competing interests: A.Z., F.Y. and D.L.
have filed a patent application based on this study.

understanding of cell tropism and pathogenesis of pestiviruses and facilitate development
of antiviral agents against CSFV.

## Introduction

Classical swine fever (CSF) is a highly contagious viral disease of swine, which causes signifi-
cant economic losses. Clinical signs of infection include fever, hemorrhage and convulsions
leading to high mortality [1]. The causative pathogen, CSF virus (CSFV), is a member of genus
*Pestivirus* within the *Flaviviridae* family and closely related to bovine viral diarrhea virus
(BVDV) and border disease virus (BDV) [2–4]. It is an enveloped virus with a single, positive-
strand RNA genome. The CSFV genome is about 12.3 kb, encoding a precursor polyprotein,
which is further processed into four structural proteins (Capsid, $E^{rns}$, E1 and E2) and eight
non-structural proteins [5].

Pestiviruses recognize the receptors on the cell surface and enter cells through receptor-
mediated endocytosis [6,7]. Inside the low-pH vesicles, fusion of the viral and cell membranes
is triggered, followed by the release of the viral genome into the cytosol [8,9]. Previous studies
have shown that CSFV enters PK15 cells via clathrin-dependent pathway [10], however the
entry of porcine alveolar macrophages is caveola-dependent [11,12].

CSFV has three envelope proteins $E^{rns}$, E1 and E2. E1 and E2 are transmembrane proteins
[5,13,14]. $E^{rns}$ and E2 are exposed on the outer layer of the viral envelope, while E1 is buried
underneath [14]. $E^{rns}$ and E2 are the major targets for neutralizing antibodies. Antibodies
against E2 can neutralize CSFV infection completely; however, those against $E^{rns}$ are only par-
tially effective [13,15,16]. E1 and E2 are sufficient to mediate CSFV entry, while $E^{rns}$ is dispens-
able, suggesting the critical role of E2 protein during viral entry [16]. Heparan sulfate and
Laminin receptor has been indicated as attachment factors for CSFV by interaction with $E^{rns}$
[17,18]. Porcine CD46 has also been shown to play a role in the initial steps of CSFV entry
[19]. However, all these factors are not essential for CSFV infection.

Herein, we identified a disintegrin and metalloproteinase-17 (ADAM17), as a CSFV attach-
ment factor by pull-down assay. ADAM17, also named tumor necrosis factor-α-converting
enzyme (TACE), is a member of the metalloproteinase superfamily and responsible for the
processing of many transmembrane proteins. It is a single-pass transmembrane protein with a
zinc-dependent metalloproteinase domain at the N-terminus. We show that CSFV E2 protein
recognizes the metalloproteinase domain to exploit ADAM17 for infection of permissive cells.
Our findings might have a significant impact on the study of the life cycle and pathogenesis of
pestiviruses.

## Results

### Identification of ADAM17 by pull-down with sE2-Fc

To identify the cellular receptor interacting with E2 protein, we expressed the extracellular
domain of E2 fused with Fc tag at the C-terminus (sE2-Fc) (Fig 1A). The sE2-Fc bound effi-
ciently with CSFV-permissive PK15 cells as measured by flow cytometry, while no binding
was detected with non-permissive BHK-21 cells (Fig 1B). The binding was completely elimi-
nated by the CSFV neutralizing antibody V3 (Fig 1B) [16]. In addition, sE2-Fc could potently
neutralize lentivirus-based CSFV pseudoparticles (CSFVpp) bearing envelope proteins from
CSFV Shimen or SXCDK strains [16,20], representing two major genotypes 1 and 2 of CSFV,
respectively (Fig 1C).

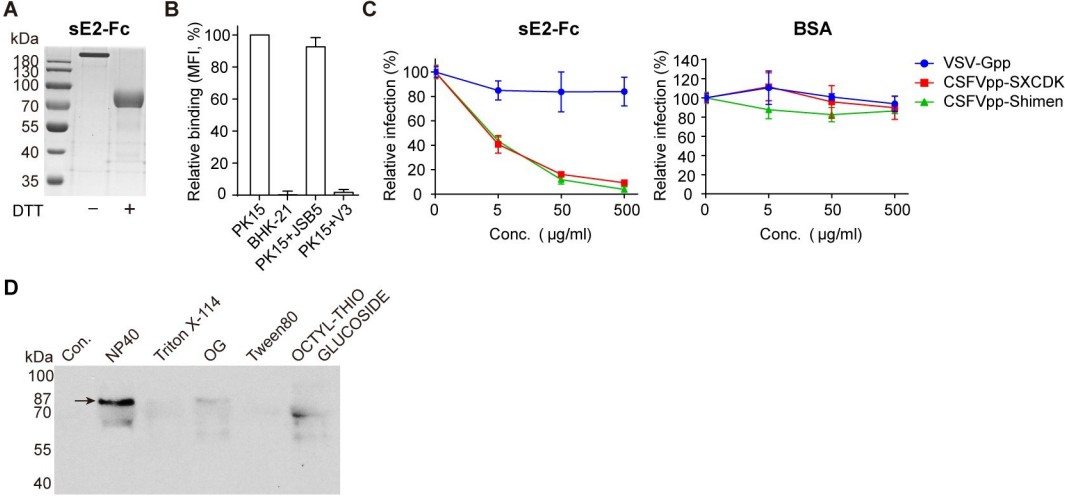

**Fig 1. Identification of ADAM17 by pull-down with sE2-Fc.** (**A**) Coomassie blue staining of sE2-Fc with or without DTT treatment. (**B**) Binding of sE2-Fc to PK15 cells, BHK-21 cells, PK15 cells in the presence of anti-integrin antibody JSB5 or anti-CSFV antibody V3 by flow cytometry analysis. Mean fluorescence intensity (MFI) values were determined using FlowJo software and normalized to MFI of PK15. (**C**) PK15 cells were pretreated with sE2-Fc (left) or BSA (right) followed by pseudoparticle infection. Error bars indicate standard deviation (SD) of mean (n = 3). The data represent three independent experiments. (**D**) Pull-down of sE2-Fc binding proteins from PK15 cell lysates solubilized by various detergents. The membrane proteins were labeled with sulfo-NHS-LC-biotin, and blotted with HRP-conjugated streptavidin.

We chose PK15 cells to provide putative prey proteins because they are highly susceptible to CSFV [16]. The membrane proteins of PK15 cells were solubilized by various detergents, and a band around 87 kDa was pulled down by sE2-Fc from the sample solubilized with NP40 (Fig 1D). Four peptides (NCQFETAQK, SPQEVKPGER, GEESTTTNYLIELIDR, FWEFIDK) were identified from the 87 kDa band by mass spectrometry matching the pig ADAM17. Human ADAM17, 92.44% identical to the porcine homolog in the amino acid sequence, is a membrane-bound metalloproteinase. The mature form of ADAM17 is ~85 kDa, which is similar in size to the band obtained from the pull-down experiments [21,22]. ADAM is a family with 40 members identified from the mammalian genomes, among which ADAM17 is the best studied [23,24].

## ADAM17 is required for CSFV entry into porcine cells

ADAM17 shares 24% amino acids identity with its closest relative ADAM10 in the ADAM family [23]. To evaluate the function of ADAM17 during CSFV entry, *ADAM17* and *ADAM10* were knocked out separately in the PK15 cells using the CRISPR-Cas9 system. sgRNAs were designed to delete the second exon of *ADAM17*, the second and third exons for *ADAM10*, resulting in frame-shifting mutations near the N-terminus, which was confirmed by Sanger sequencing (S1 Fig). The binding of sE2-Fc protein with the PK15 *ADAM17*-KO (knockout) cells was completely eliminated as shown by flow cytometry, while it was retained in *ADAM10*-KO cells (Fig 2A). Compensation of ADAM17 by stable expression of pig *ADAM17* (p*ADAM17*) cDNA in *ADAM17*-KO cells fully rescued sE2-Fc binding (Fig 2A). These results suggest that ADAM17 is critical for E2 binding with CSFV-permissive cells.

The involvement of ADAM17 in CSFV entry was further investigated using both CSFVpp and authentic cell culture grown CSFV (CSFVcc). Consistent with the binding results, loss of *ADAM17* blocked the CSFVpp entry as well as CSFVcc infection (Fig 2B and 2C). In contrast, no effects were observed when *ADAM10* was absent (Fig 2B and 2C). Reintroduction of pig

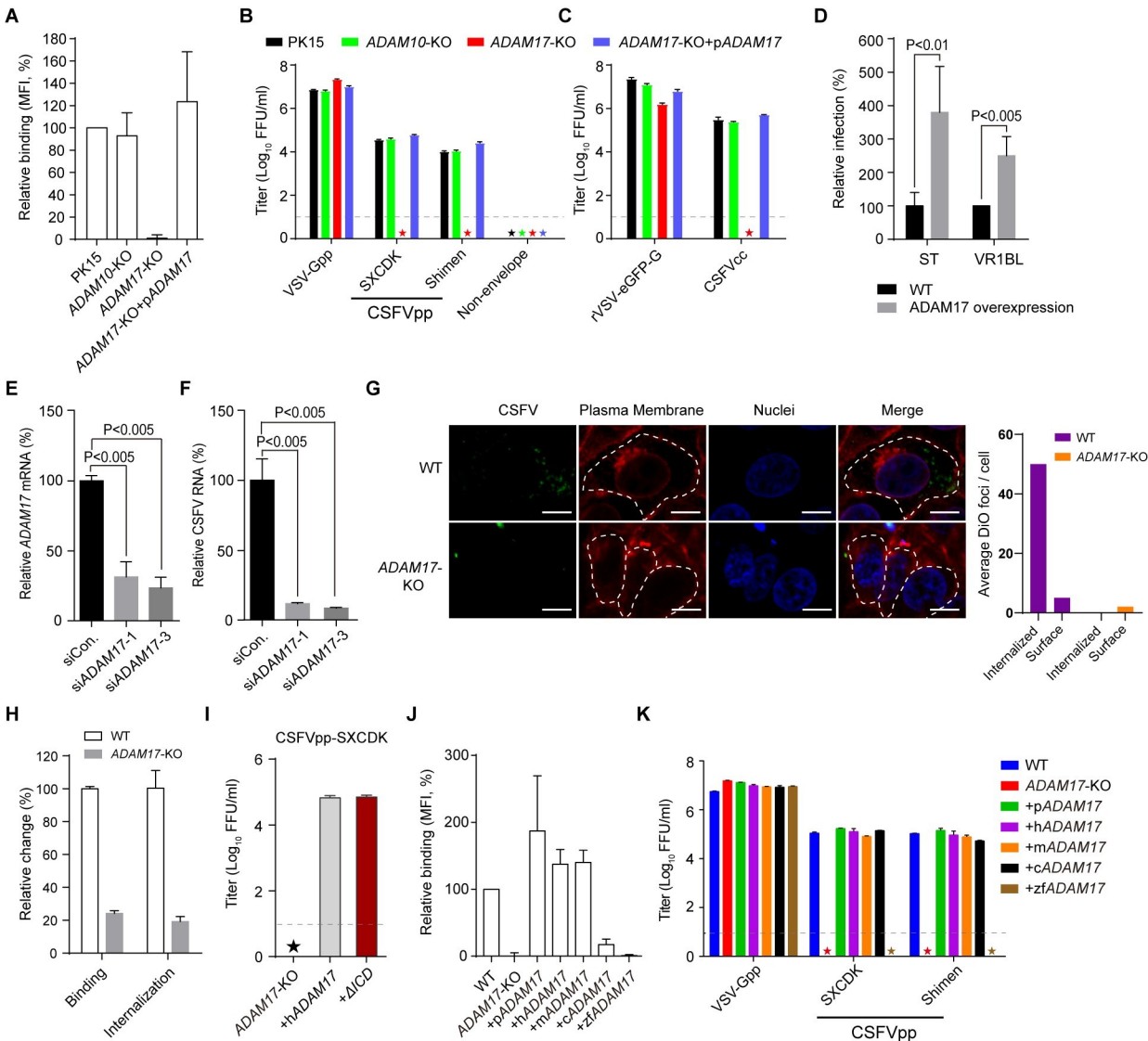

**Fig 2. ADAM17 is required for CSFV entry into porcine cells.** (**A**) Binding of sE2-Fc to PK15, PK15 *ADAM10* knockout (*ADAM10*-KO), PK15 *ADAM17* knockout (*ADAM17*-KO), or pig *ADAM17* trans-complemented *ADAM17*-KO cells (*ADAM17*-KO+p*ADAM17*) by flow cytometry. (**B**) Cell lines indicated above were infected with lentiviruses encoding a GFP reporter and pseudotyped with envelope proteins from CSFV SXCDK (CSFVpp-SXCDK) or Shimen (CSFVpp-Shimen) strains. The infection titer was measured as focus-forming units per ml (FFU/ml). Pseudoparticles enveloped with VSV G protein (VSV-Gpp) served as the positive control, and that bearing no envelope protein (Non-envelope) was the negative control. (**C**) Indicated cells were infected with CSFVcc C strain or recombinant VSV encoding GFP reporter (rVSV-eGFP-G). Viral infectivity was expressed as E2 or GFP-positive foci. (**D**) Relative CSFVpp-Shimen infection of poorly permissive porcine cell lines with or without ADAM17 overexpression. (**E-F**) Porcine primary embryonic fibroblasts were infected with CSFVcc 60 h after transfection with irrelevant (si*Con.*), or *ADAM17*-specific siRNAs (si*ADAM17*-1 and si*ADAM17*-3). *ADAM17* mRNA levels (**E**) and CSFV RNA in the supernatants (**F**) were measured by quantitative RT-PCR (qRT-PCR) and normalized to siCon. treatment. (**G**) DiO-labeled CSFVcc (green) was incubated with PK15 WT or *ADAM17*-KO cells on ice and then warmed to 37°C, allowing internalization. The plasma membrane was stained with Alexa Fluor 594 conjugated wheat germ agglutinin (red) and the nuclei were stained with Hoechst 33342 (blue). Size bars indicate 20 μm. The number of DiO foci was counted in about 100 cells. (**H**) PK15 WT or *ADAM17*-KO cells were incubated with purified CSFVcc at 4°C or 37°C as described in the Materials and Methods. Cells were collected and RNA (CSFV and β-actin) was measured by qRT-PCR. (**I**) CSFVpp infection in *ADAM17*-KO cells stably expressing h*ADAM17* or h*ADAM17 ΔICD* (deletion of intracellular domain). (**J**) Flow cytometry analysis of sE2-Fc binding with PK15 (WT), *ADAM17*-KO and *ADAM17*-KO trans-complemented with *ADAM17* from pig (+p*ADAM17*), human (+h*ADAM17*), mouse (+m*ADAM17*), chicken (+c*ADAM17*), and zebrafish (+zf*ADAM17*). (**K**) Cells were infected with indicated pseudoparticles and the titer was measured as FFU/ml. In (**B**) (**C**) (**I**) (K), the lower limit of detection was 10 FFU/ml (dashed line). Asterisks indicate samples below the limit. In (**D-F**), significance was calculated using a multiple *t* test and *P* values were showed. Error bars indicate standard deviation (SD) of mean (n = 3). The data represent three independent experiments.

*ADAM17* cDNA into the *ADAM17*-KO cells completely rescued these phenotypes (Fig 2B and 2C). The control VSV-Gpp and rVSV-eGFP-G infected all the cells at similar levels (Fig 2B and 2C). The compensation of porcine *ADAM17* in PK15 *ADAM17*-KO cell lines was confirmed by immunofluorescence using an antibody against human ADAM17, which also cross-reacts with over-expressed porcine and murine homologs, but not with endogenous ADAM17s (S2A Fig). In addition, overexpression of ADAM17 in two poorly permissive cell lines: ST (swine testicle cell line) and VR1BL (porcine fetal retina cell line) significantly enhanced the susceptibility to CSFVpp (Fig 2D).

To investigate the role of ADAM17 in mediating CSFV entry in primary cells, *ADAM17* was knocked down in primary porcine embryonic fibroblasts (PEFs) using two *ADAM17*-specific siRNAs. As shown in Fig 2E, *ADAM17* mRNA levels decreased by 70.9% and 78.9%, respectively. Correspondingly, CSFV RNA in CSFVcc infected *ADAM17*-knockdown PEFs measured by qRT-PCR was significantly decreased by 89.2% and 92.4% (Fig 2F), indicating that ADAM17 is critical for CSFV infection in primary cells.

## ADAM17 is responsible for CSFV virion attachment

CSFV attaches to one or more cell-surface molecules and is internalized via clathrin-dependent endocytosis. An internalization assay was performed to visualize CSFV entry in PK15 and *ADAM17*-KO cells. Pre-bound DiO-labeled CSFVcc was internalized into PK15 cells efficiently upon 37˚C treatment (Fig 2G). However, only a few DiO foci were observed in the *ADAM17*-KO cells, mostly on the plasma membrane, probably due to the interaction through E$^{rns}$ (Fig 2G). In order to dissect the role of ADAM17 in CSFV entry, we performed the binding and internalization assays. Both binding and internalization were reduced in *ADAM17*-KO cells, at a similar ratio (4.1 and 5.2 fold) as determined by qRT-PCR (Fig 2H). To further investigate the role of ADAM17 in CSFV internalization, we deleted the intracellular domain, which mediated ADAM17 internalization via the interaction with host protein PACS-2 [25]. The ADAM17 mutant without the intracellular domain (+*ΔICD*) was able to fully restore CSFVpp infectivity in *ADAM17*-KO cells (Fig 2I), suggesting the decrease of internalization in Fig 2H was probably due to the reduction of viral attachment. These data demonstrated that ADAM17 serves as an attachment factor during CSFV entry.

## ADAM17 is not a host determinant of CSFV

CSFV displays very narrow host tropism. It only infects domestic pigs and wild boars [26]. Similarly, CSFVpp does not infect human and mouse cell lines [16]. The ADAM17s of mouse, human, chicken and zebrafish (mADAM17, hADAM17, cADAM17 and zfADAM17) share 89.68%, 92.44%, 74.91% and 58.55% identities in amino acid sequences with pig ADAM17 as determined by BLAST. To address whether ADAM17 is the determinant of the host-tropism of CSFV, we overexpressed the ADAM17s from these four species in *ADAM17*-KO cells. Expression of m*ADAM17* and h*ADAM17* was confirmed by immunofluorescence using an anti-hADAM17 polyclonal antibody (S2A Fig). In order to detect the expression of cADAM17 and zfADAM17, which could not be recognized by anti-hADAM17 antibody, we inserted a flag tag at the C-terminal of p*ADAM17* (p*ADAM17*-F), cADAM17 and zfADAM17. Expression of flag-tagged ADAM17s was detected by western blot using an anti-flag antibody (S2C Fig). The flag tag did not affect the CSFVpp infectivity (S2B Fig). The eliminated sE2-Fc binding in *ADAM17*-KO cells was restored by reintroduction of human and mouse *ADAM17* at similar levels as pig *ADAM17* (Fig 2J). In contrast, the distant homologs cADAM17 and zfADAM17 displayed much lower binding efficiency with sE2-Fc, at ratios of 20% and zero respectively as compared with pADAM17 (Fig 2J). Correspondingly, mouse, human and

chicken ADAM17 could confer permissiveness of CSFVpp to *ADAM17*-KO cells as efficient as pig ADAM17 (Fig 2K). However, ectopic expression of zfADAM17 in *ADAM17*-KO cells resulted in almost no CSFVpp entry (Fig 2K). Therefore, ADAM17 is not a determinant of species host range, suggesting that other entry factors defining the host tropism remain to be discovered.

## ADAM17 expression correlates with CSFV susceptibility

ADAM17 is ubiquitously expressed, which is consistent with the broad cell tropism of CSFV [24,27]. To validate whether ADAM17 expression correlates with CSFV susceptibility, we surveyed the CSFV permissiveness and the *ADAM17* mRNA levels on a panel of porcine cell lines. Among the four cell lines tested, PK15, IPEC-J2 (intestinal porcine enterocytes) and 3D4/21 (porcine monomyeloid cell line) were highly permissive, whereas the entry of CSFVpp was 2.5–3 logs lower in ST cells (Fig 3A). Interestingly, 3D4/21 cells, with 49% ADAM17 mRNA as compared with PK15, were still highly susceptible to CSFVpp, whereas ST cells (with 37% ADAM17) were poorly permissive (Fig 3B). We then investigated the mRNA levels of TIMP-3, a native tissue inhibitor of ADAM17, which could bind to the metalloproteinase domain of ADAM17 [28]. The *TIMP-3* expression level in ST cells was three-fold higher than that in PK15 cells (Fig 3B). Thus, the ratio of *TIMP-3* mRNA to *ADAM17* in ST cells was about 8.8-fold higher than that of PK15 cells, followed by 2.1- and 1.5-fold than that of 3D4/21

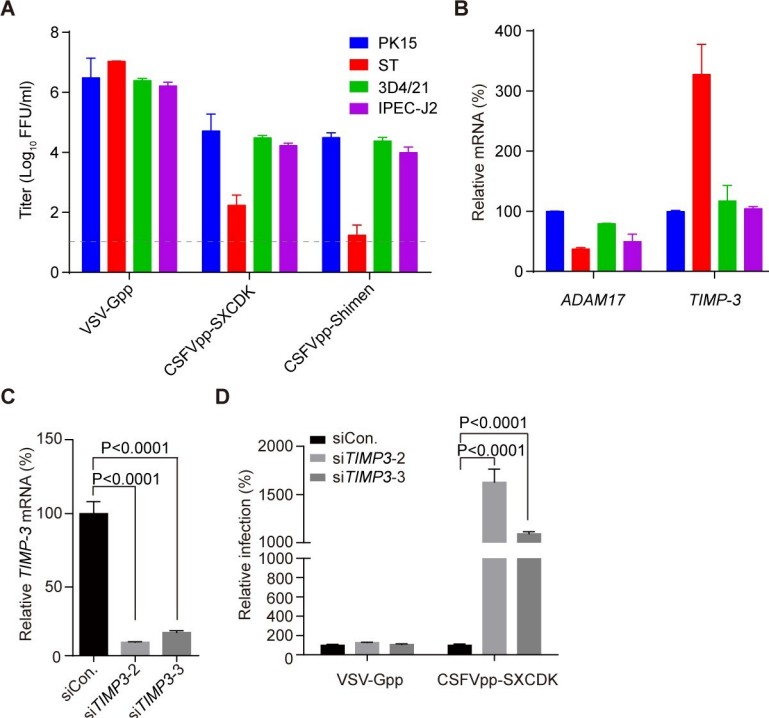

**Fig 3. ADAM17 expression correlates with CSFV susceptibility.** (**A**) PK15, ST, 3D4/21 and IPEC-J2cells were infected with the indicated pseudoparticles. (**B**) The expression levels of *ADAM17* and *TIMP-3* were measured by qRT-PCR. The mRNA level in PK15 cells was set as 100%. (**C**) Silencing of *TIMP-3* in ST cells with two oligos (si*TIMP3*-2 and si*TIMP3*-3). *TIMP-3* mRNA levels were measured by qRT-PCR and expressed as a percentage of control oligo (siCon.). (**D**) CSFVpp-SXCDK and VSV-Gpp infection of siRNA-treated ST cells normalized to siCon.-treated cells. Error bars indicate standard deviation (SD) of the mean (n = 3). Significance was calculated using a multiple *t* test. The data represent three independent experiments.

and IPEC-J2 cells respectively. Knock-down of *TIMP-3* in ST cells by siRNAs enhanced susceptibility to CSFVpp, suggesting that the low permissiveness of CSFVpp in ST cells was due to the occupancy of ADAM17 by the high level of TIMP-3 (Fig 3C and 3D). These results indicate that ADAM17 expression correlates with CSFV susceptibility.

## CSFV interacts with ADAM17 through the metalloproteinase domain

The extracellular region of mature ADAM17 contains a pro-domain, a metalloproteinase domain, a disintegrin domain and a cysteine-rich domain. Since TIMP-3 binds to ADAM17 through the metalloproteinase domain [28,29], we propose that CSFV E2 engages ADAM17 by binding to the same domain. ADAM17 is a zinc-dependent metalloproteinase with one zinc ion in the active site of the metalloproteinase domain (S4 Fig) [30]. The 1,10-phenanthroline can inhibit ADAM17 activity by chelating the zinc ion [31]. After pre-treating PK15 cells with 1, 10-phenanthroline, sE2-Fc binding was reduced in a dose-dependent manner with a 74% decrease at 25 mM (Fig 4A). Zinc ion is coordinated with the HEXGHXXGXXHD motif in the metalloproteinase domain and replacement of the H405 with aspartic acid can eliminate zinc binding (S3 Fig) [32]. To further investigate involvement of the metalloproteinase domain in CSFV entry, H405D mutation was introduced into h*ADAM17* and then stably transduced into *ADAM17*-KO cells. The h*ADAM17* H405D mutant failed to restore sE2-Fc binding as well as CSFVpp permissiveness in the *ADAM17*-KO cells (Figs 4B, 4C and S2A).

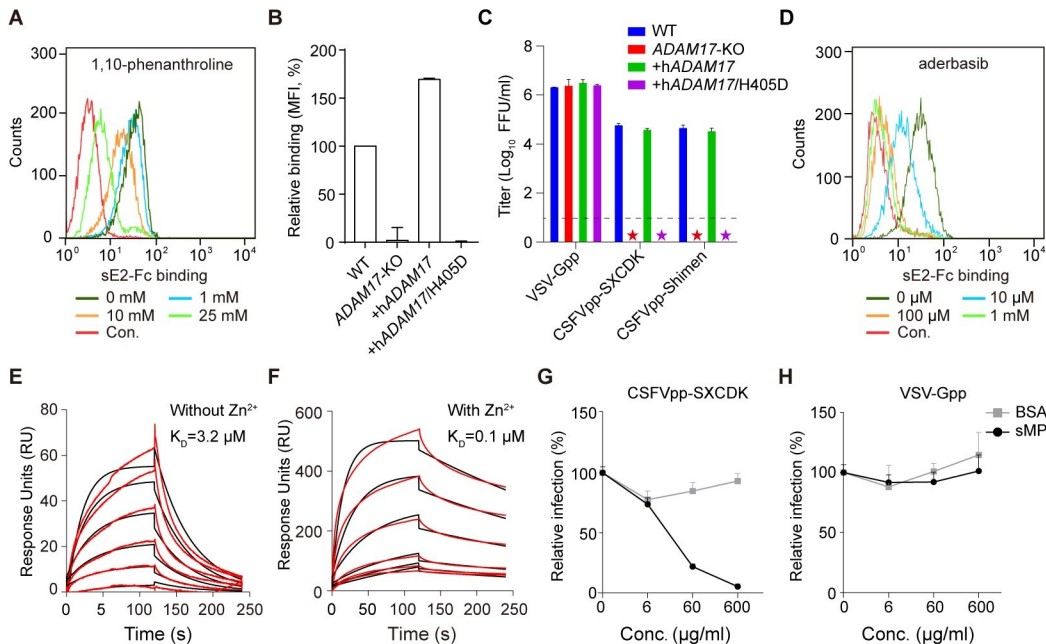

**Fig 4. E2 binds the metalloproteinase domain of ADAM17.** (**A**) PK15 cells were treated with indicated concentrations of the ion chelator: 1, 10-phenanthroline, and sE2-Fc binding was tested by flow cytometry analysis. *ADAM17*-KO cells transduced with h*ADAM17* WT or H405D mutant were processed for sE2-Fc binding (**B**) and pseudovirus infection (**C**). The lower limit of detection was 10 FFU/ml (dashed line). Asterisks indicate samples below the limit. (**D**) PK15 cells were treated with indicated concentrations of aderbasib, and sE2-Fc binding was determined by flow cytometry analysis. Binding between soluble ADAM17 metalloproteinase domain (sMP) and sE2 was characterized by surface plasmon resonance (SPR) with (**F**) or without (**E**) zinc ions in the running buffer. Experimental curves (red lines) were fit using a 1:1 binding model (black lines). The equilibrium dissociation constant ($K_D$) were calculated by BIAcore T100 Evaluation software. CSFVpp (**G**) and VSV-Gpp (H) infection of PK15 cells pretreated with sMP or BSA. Error bars indicate SD of the mean (n = 3). The above data represent three independent experiments.

A class of ADAM17 inhibitors including aderbasib inhibits the metalloprotease activity through binding to the active site of the metalloproteinase domain (S4 Fig). Aderbasib (INCB007839), is a selective inhibitor of ADAM10 and ADAM17, which is currently in clinical trials for cancer treatment [33]. To evaluate whether aderbasib could alter the interaction between ADAM17 and sE2-Fc, we incubated the trypsinized PK15 cells with different concentrations of aderbasib and then analyzed binding of ADAM17 to sE2-Fc by flow cytometry analysis. As the concentration of the compound increased, binding of sE2-Fc decreased accordingly, with almost no binding detected at 100 μM (Fig 4D). Similarly, aderbasib showed antiviral effect against CSFV pseudovirus at 100 μM and 1 mM (S5A and S5B Fig). Together, these results suggest that the interaction site between E2 and ADAM17 is located in the metalloproteinase domain of ADAM17.

To determine whether ADAM17 directly binds to E2, we expressed and purified the soluble metalloproteinase domain of pig ADAM17 (sMP) and E2 protein (sE2) with the $(his)_6$ tag (S2D Fig). The binding affinity between sMP and sE2 was 3.2 μM without zinc and enhanced to 0.1 μM by addition of zinc as analyzed by surface plasmon resonance (SPR) (Fig 4E and 4F). Pre-incubation with sMP, but not the control BSA, reduced CSFVpp susceptibility in a concentration-dependent manner (Fig 4G). No effect was detected with VSV-Gpp (Fig 4H). These results demonstrated that CSFV E2 directly recognize ADAM17 through its metalloproteinase domain.

## Key region in ADAM17 responsible for CSFV virion attachment

To dissect the key region in ADAM17 responsible for CSFV virion attachment, the sequences of zf*ADAM17* metalloproteinase domain were substituted with those of p*ADAM17* (Fig 5A). Replacement of the entire metalloproteinase domain of zf*ADAM17* with that of p*ADAM17* completely restored sE2-Fc binding and CSFVpp entry. We substituted parts of the zf*ADAM17* metalloproteinase domain with counterparts of p*ADAM17*, such as aa301-427, aa301-345 and aa346-427 (Fig 5A). The expression of chimeric proteins was detected by an anti-Flag antibody recognizing the flag tag fused to their C-terminus (Fig 5D). Both p301-427 and p301-345 restored sE2-Fc binding, while almost no improvement was observed for p346-427 (Fig 5B). Consistent with the binding data, p301-427 and p301-345 rendered CSFVpp susceptibility as efficiently as p*ADAM17* (Fig 5C). Therefore, the aa301-345 in the metalloproteinase domain appears to play a critical role in CSFV virion attachment.

## Discussion

ADAM17 is a membrane-bound metalloproteinase consisting of a pro-domain, a metalloproteinase domain, a disintegrin domain, cysteine-rich region, a transmembrane domain, and a cytoplasmic domain [21,24]. The pro-domain is cleaved by furin during maturation. Pig ADAM17 shares 92.44% amino acid identity with its human homolog, which is 85 kDa after maturation [21,22]. Cellular localization and tissue distribution are important for a protein to serve as a virus entry factor. Although most of the mature ADAM17 localizes in the prenuclear region, a small amount is translocated to the plasma membrane [24], which is accessible for CSFV. ADAM17 is widely expressed in almost every tissue, with very high levels in heart, placenta, skeletal muscle, pancreas, spleen, thymus, prostate, testes, ovary, and small intestine [21]. This is consistent with the systemic infection of CSFV in pigs. Moreover, high levels of *ADAM17* expression were detected in all of the CSFV-permissive cells tested. Thus, ADAM17 has the potential to serve as a functional entry factor for CSFV.

ADAM17 belongs to a large family of more than 40 members in mammals. There is very little amino acid sequence similarity between ADAM17 and other ADAMs [23]. ADAM10 is the

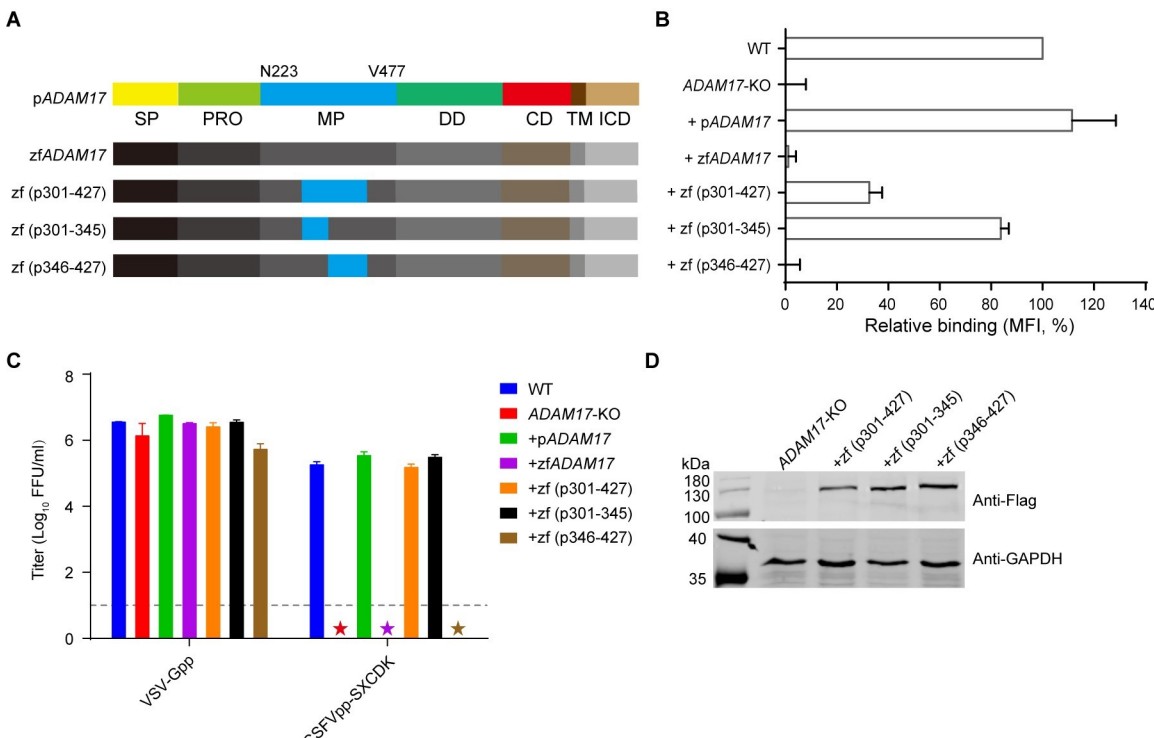

**Fig 5. Key region in ADAM17 responsible for CSFV entry.** (**A**) Schematic diagram of p*ADAM17*, zf*ADAM17* and construction of the chimera *ADAM17*s. Chimera *ADAM17*s were constructed by replacing the indicated fragments of zf*ADAM17* with those of p*ADAM17*. SP, signal peptide; PRO, pro-domain; MP, metalloproteinase domain (N223-V477); DD, disintegrin domain; CD, catalytic domain; TM, transmembrane domain; ICD, intracellular domain. (**B**) Relative binding of sE2-Fc with PK15 (WT), *ADAM17*-KO and *ADAM17*-KO stably transduced with p*ADAM17* (+p*ADAM17*), zf*ADAM17* (+zf*ADAM17*) or various *ADAM17* chimeras by flow cytometry analysis. (**C**) Above cell lines were infected with indicated pseudoparticles. The lower limit of detection was 10 FFU/ml (dashed line). Asterisks indicate samples below the limit. (**D**) Western blotting of chimera ADAM17s in the cell lysates using anti-Flag antibody, with anti-GAPDH antibody as the control. Error bars in (B) and (C) indicate SD of the mean (n = 3). The data represent three independent experiments.

closest homolog of ADAM17 and only shares 24% sequence identity with ADAM17. In the highly CSFV-permissive PK15 cell line, the expression of both ADAM17 and ADAM10 was detected. Knockout of *ADAM17* in PK15 cells completely abolished CSFV entry, while no effect was seen when *ADAM10* was deficient. Considering most of the ADAMs are expressed in restricted tissues [34], we speculate that ADAM17 is the only, or at least the major ADAM family member related to CSFV entry.

*Flaviviridae* entry is complicated and usually involves multiple entry factors. For example, entry of HCV, best known among the *Flaviviridae* family, is mediated by a series of entry factors including CD81, SR-BI, CLDN1 and OCLDN [35–40]. Some TIM- and TAM- family members also play roles in the attachment of flaviviruses and pestiviruses [41,42]. Our studies revealed that CSFV envelope protein E2 directly recognized the metalloproteinase domain to exploit ADAM17 on the plasma membrane to initiate virion attachment. Furthermore, ADAM17 expression pattern correlates with the broad tissue tropism of CSFV in pigs [27]. In addition, ADAM17 homologs from unsusceptible mammals also conferred CSFV permissiveness, suggesting other co-factors are needed to define the CSFV host tropism.

Entry factors are promising targets for antiviral therapy and development of pathogen-resistant domestic animals. PPRSV is another important pig pathogen that uses CD163 as a cellular receptor [43]. PPRSV-resistant pigs have been successfully developed by editing the

viral binding epitope in CD163 [44]. Our results demonstrate that aa301-345 in the metallo-proteinase domain is involved in virus-host recognition. Crystal structure of human ADAM17 reveals that aa301-345 contains a loop, an α-helix and a β-strand forming an exposed grove, which provides a potential viral binding site (S4 Fig) [45]. This region is adjacent to the metalloproteinase active center and loss of zinc or addition of aderbasib might block E2 interaction by affecting its conformation. Furthermore, the sequences of pig, human and mouse are identical in this region, whereas the zebrafish homolog only shares 64% amino acid identity with them. Among the 15 distinct amino acids in this region in zfADAM17, 10 are differently charged, which might explain the failure of E2 protein interaction. Therefore, future studies will attempt to identify the key residues involved in the interaction between CSFV E2 and ADAM17 by structural studies or mutagenesis. Residues involved in viral recognition without affecting ADAM17 physiological function are potential gene editing targets for CSFV-resistant pig development.

## Materials and methods

### Cell lines, viruses and antibodies

Porcine cell lines PK15 (CCL-33), ST (CRL-1746) and 3D4/21 (CRL-2843), human cell line HEK293T (CRL-3216) and hamster cell line BHK-21 (CCL-10) were obtained from ATCC. IPEC-J2 (MZ-0703) was obtained from MINGZHOUBIO (Ningbo, China) and VR1BL (porcine fetal retina cell line) was provided by Dr. Enqi Du (College of Veterinary Medicine, Northwest A&F University). Above cells were grown in Dulbecco Modified Eagle medium (DMEM) with 10% fetal bovine serum (FBS), 1% L-glutamine, and 1% penicillin-streptomycin. Pig embryonic fibroblasts (PEF) were provided by Dr. Jianguo Zhao (Institute of Zoology, Chinese Academy of Sciences) and obtained from embryonic day 35 Bama miniature pig embryos, grown in DMEM containing 15% FBS, 1% non-essential amino acids, 1% L-glutamine and 1% penicillin-streptomycin. All cells were incubated at 37°C with 5% $CO_2$.

The rVSV-eGFP-G was a gift from Dr. Kartik Chandran (Albert Einstein College of Medicine) [46]. The CSFV C strain was purchased from Weike Biotec, Harbin, China.

JSB5 is a mouse monoclonal antibody (mAb) targeting integrin α5β1 (Chemicon International, Harrow, UK). V3 is a mouse mAb against CSFV E2 that has been discontinued (CEDI-Diagnostics, Lelystad, Netherlands). WH303 is a mouse mAb targeting CSFV E2 (RAE0826, APHA Scientific, UK). Anti-human ADAM17 is a rabbit polyclonal antibody purchased from Abcam (ab39162, Cambridge, UK). Anti-GAPDH mouse mAb was purchased from Beijing ComWin Biotech Co.,Ltd. (CW0100, Beijing, China). Anti-Flag mouse mAb was purchased from Thermo Fisher Scientific (MA1-91878, Waltham, MA, USA).

### Plasmids construction

Pig *ADAM17* (Genebank: NM_00109926.1) was amplified from the cDNA of PK15 cells, and human *ADAM17* (Genebank: NM_003183.5) was cloned from a PRK5F-TACE plasmid, which was a gift from Rik Derynck (University of California, San Francisco) [47]. Mouse, chicken and zebrafish *ADAM17*s (Genebank: NM_009615.6, AY486557.1, XM_021468637.1) were synthesized by Beijing Shengyuan Kemeng Gene Biotechnology Co., Ltd. These *ADAM17* homologs were cloned into the lentivirus vector by Gibson assembly. To generate CSFV pseudoparticles for entry experiments, plasmids expressing CSFV envelope proteins were constructed by inserting human codon-optimized DNA fragments encoding $E^{rns}E1E2$ of CSFV genotype 1 Shimen strain (GenBank:AF333000.1) or genotype 2 SXCDK strain (Genbank:GQ923951.1) (synthesized by Beijing Shengyuan Kemeng Gene Biotechnology) into a

pCAGGS vector between XbaI and EcoRI sites (referred to as pCAGGS-CSFV/Shimen-E012 and pCAGGS-CSFV/SXCDK-E012) [16].

### Stable cell lines construction

To construct *ADAM17* over-expressing cell lines, lentiviruses were packaged by transfecting HEK293T cells in 10 cm plates with 12.5 μg lentivirus plasmids carrying *ADAM17* cDNAs, 7.5 μg psPAX2 and 5 μg pMD2.G using calcium phosphate method. The supernatant was collected 48 h post-transfection and used for infection of the target cells. After infection for 48 h, the cells were selected with 3 μg/ml puromycin for 7 d.

### Protein expression and purification

The ectodomain of CSFV E2 (sE2-Fc) encoding 690–1020aa of the polyprotein was amplified from pCAGGS-CSFV/Shimen-E012 by PCR and inserted into pPUR-TPA-Fc between BamHI and EcoRI with a TPA signal peptide at the N-terminus and a Fc tag at the C-terminus. The plasmid, referred to as pPUR-TPA-CSFV-sE2-Fc, was transfected into HEK293T cells by poly-ethylenimine (PEI, Polysciences, Warrington, PA, USA). At 12 h post-transfection, the original medium was replaced with DMEM plus 2% FBS. Supernatants were harvested 36 h later and centrifuged at 5000 g for 10 min to remove cell debris. The sE2-Fc protein was purified first by protein A affinity chromatography (GE Healthcare, Chicago, IL, USA), and then by gel filtration on a HiLoad 16/600 Superdex 200 pg column (GE Healthcare, Chicago, IL, USA).

The coding sequence for CSFV sE2 (693-1020aa) was cloned into the pFastBac1 vector (Invitrogen, Carlsbad, CA, USA). The sequence encoding the pro-domain and metalloproteinase domain of pig ADAM17 (18-477aa, GenBank: NM_001099926.1) was inserted into pFast-Bac-dual vector (Invitrogen, Carlsbad, CA, USA) with two mutations (S266A and N452Q) to prevent N-linked glycosylation. To facilitate protein purification, a gp67 signal peptide and a $(His)_6$ tag were added to both constructs at the N-terminus and C-terminus respectively. The proteins, designated as sE2 and sMP, were expressed using the Bac-to-Bac baculovirus expression system (Invitrogen, Carlsbad, CA, USA) and purified as described in Wang et al., 2016 [48].

### Pull-down and mass spectrometry

PK15 cells were labeled using membrane-impermeable sulfo-NHS-LC-biotin (Pierce, Rockford, IL, USA). Briefly, PK15 cells in 10 cm plates were washed with PBS (pH 8.0) twice and then labeled with sulfo-NHS-LC-biotin diluted in PBS at 0.5 mg/ml for 30 min on ice. After stopping the reaction by adding 100 mM glycine, the cells were scraped and lysed in TBS buffer (50 mM Tris-Cl, pH 7.5, 150 mM NaCl) plus 1% NP40 (Sigma-Aldrich, Germany) on ice for 1 h. One milliliter of cell lysate was cleared by centrifugation at 12,000 g for 15 min and pulled down with sE2-Fc at 5 μg/ml plus 10 μl of protein A sepharose beads (GE Healthcare, Chicago, IL, USA). After rocking overnight at 4˚C, the beads were washed four times with lysis buffer and then eluted with 50 mM glycine at pH 3.0. The eluate was separated by SDS-PAGE and stained with HRP-conjugated streptavidin (Thermo Fisher Scientific, Waltham, MA, USA).

For mass spectrometry (MS) analysis, PK15 cells from ten 25 cm plates were scraped, and the cell membranes were purified. The cell membranes were lysed with TBS plus 1% NP40 and then pulled down with sE2-Fc as described above. The eluate was separated by SDS-PAGE and stained with Coomassie blue. Bands on the gel were sliced for MS analysis at the National Center of Biomedical Analysis, Beijing, China.

## Pseudoparticles preparation and entry assay

The pseudoparticles were packaged by transfecting HEK293T cells in 10 cm dishes with 3 μg pCAGGS-CSFV/Shimen-E012, pCAGGS-CSFV/SXCDK-E012, or VSV-G, 4.8 μg of pLP1, 4.2 μg of pLP2 and 12 μg of pCDH-CMV-eGFP-IRE3-PURO encoding an eGFP reporter using the calcium phosphate method [16]. Pseudoparticles in the supernatants were harvested 48 h later and centrifuged at 12,000 g for 10 min to remove the cell debris. Then the supernatant was used in entry assays.

To perform the pseudoparticle entry assay, cells were seeded in 96 well plates at 8000 cells per well 24 h prior to infection. At 48 h after infection by pseudoparticles (100 μl/well of 10-fold serial dilutions), the titers were determined by counting GFP-positive foci (unit: FFU/ml). The detection limit of this assay was 10 FFU/ml.

## Inhibition assay

Serially diluted proteins or drugs were incubated with 100 FFU of pseudoparticles at room temperature for 30 min and then layered onto the PK15 cells in 96 well plates. Fresh media was changed 2 h after infection and the titers were determined 48 h later as described above.

## Virus titration

CSFVcc titers were determined in 96 well plates by a focus-forming assay. Cells were seeded at 8000 cells per well 24 h prior to infection. Cells were infected with 100 μl/well of 10-fold serial dilutions of viruses for 3 h at 37˚C, and then the supernatant was replaced with fresh media plus 20 mM $NH_4Cl$. After 48 h, the cells were fixed with cold methanol and stained with the mouse mAb WH303 recognizing CSFV E2 protein at a dilution of 1:500. After incubation with Alexa-Fluor 690 goat anti-mouse IgG (Thermo Fisher Scientific, Waltham, MA, USA), the viral titers were determined using Nikon Ti2 microscope.

## Flow cytometry analysis

The cells were trypsinized and resuspended in PBS supplemented with 2% FBS. Aliquots of cells were incubated with 20 μg/ml CSFV sE2-Fc for 1 h on ice followed by three washes with cold PBS plus 2% FBS. Then the cells were incubated with Alexa-Fluor 488-conjugated goat anti-human IgG (Thermo Fisher Scientific, Waltham, MA, USA) at a dilution of 1:200 on ice for 30 min followed by three washes with the cold PBS supplemented with 2% FBS. Cells were resuspended in 500 μl PBS supplemented with 2% FBS and analyzed using a BD FACSCalibur (BD Biosciences, San Jose, CA, USA).

## Immunofluorescence

Cells were cultured in 96 well plates at 12,000 cells per well. After 24 h, the cells were washed twice with PBS (pH 8.0) and fixed with methanol for 30 min at -20˚C. Subsequently, cells were blocked with 5% bovine serum albumin (BSA) in PBS at room temperature for 1 h. To detect ADAM17 protein, cells were incubated with rabbit anti-hADAM17 (1:200) overnight at 4˚C. After being washed with PBS three times, cells were incubated with Alexa Fluor 488-conjugated goat anti-rabbit IgG (1:500) (Thermo Fisher Scientific, Waltham, MA, USA) for 1 h at room temperature. Hoechst 33342 (Thermo Fisher Scientific, Waltham, MA, USA) was added at 1 μg/ml to stain the nuclei. Representative images were captured with a Nikon Ti2 microscope.

## CRISPR-Cas9 knockout

SgRNAs were designed using Cas-Designer (http://www.rgenome.net/cas-designer/). Two sgRNAs located in the introns were designed to delete one or two exons, resulting in frameshift mutations. The sgRNA-up was located upstream of the exon, and sgRNA-down was located downstream. They were synthesized by Generay Biotech (Shanghai, China) and separately ligated into a PX330 vector using a BbsI restriction site. To construct the knockout cell lines, PK15 cells in 10 cm dishes were co-transfected with 12.5 μg of PX330-sgRNA-up together with 12.5 μg PX330-sgRNA-down using 60 μl of FuGENE 6 (Promega, Fitchburg, WI, USA) transfection reagent. After 36 h, the individual GFP-positive cells were sorted into 96 well plates by Beckman MoFlo XDP (Beckman Coulter, Brea, CA, USA). Positive cell clones were screened by genomic PCR using primers flanking the target exons.

## RNAi

The siRNAs were designed using BLOCK-iT RNAi Designer (http://rnaidesigner. thermofisher.com/rnaiexpress/) and synthesized by Sangon Biotech (Shanghai, China). The ADAM17-specific siRNAs were: 5'-GCAACAAAGUGUGUGGCAA-3' (si*ADAM17*-1) and 5'-GGUGUUUGUCCAAAGGCUU-3' (si*ADAM17*-3). The TIMP3-specific siRNAs were: 5'-GCUACUACCUGCCUUGCUU-3' (si*TIMP3*-2) and 5'-GCCGUGUCUAUGAUGGCAA-3' (si*TIMP3*-3). The non-targeting siRNA (5'-UUCUCCGAACGUGUCACGU-3') was used as an irrelevant control (siCon.).Cells in 6 cm dishes at 95% confluence were transfected with 50 nM siRNA using 18 μl Lipofectamine RNAimax (Invitrogen, Carlsbad, CA, USA). Total RNAs were extracted 36 h later to test the interference efficacy by qRT-PCR. For virus infection, cells were seeded into 96 well plates 36 h post-transfection and infected 24 h later.

## Quantitative RT-PCR

Cells were lysed in TRIzol (Invitrogen, Carlsbad, CA, USA) and RNA was extracted according to the manufacturer's protocol. Quantification of gene expression or viral genomic copies was performed using One Step TB Green PrimeScript RT-PCR Kit (TaKaRa, Japan) on Applied Biosystems QuantStudio (Thermo Fisher Scientific, USA). Each sample was measured in triplicate. Primers used for qRT-PCR were: 5'-GCTTGGTTCCTATCGTGCTG-3' (ADAM17-F) and 5'-TGGCGAATGCTGGATAAAGA-3' (ADAM17-R); 5'-GCTGACAGGCCGTGTCTATGAT-3' (TIMP3-F) and 5'-CAAGGCAGGTAGTAGCAGGATTT-3' (TIMP3-R); 5'-GAAGGGTA GTCGGCAGGGTCA-3' (CSFV-F); 5'-CGGCACCTGTAGCAAGGGTTAT-3' (CSFV-R).

## Surface plasmon resonance (SPR)

SPR analysis was performed on Biacore T100 system (GE Healthcare). CSFV sE2 protein (20μg/ml) was immobilized on the CM5 sensor chip (GE Healthcare, Chicago, IL, USA) using a standard amine coupling method. A blank channel was used as the negative control. Different concentrations of sMP protein (5 μM, 2.5 μM, 1.25 μM, 0.625 μM, 0.312 μM, 0.156 μM) were then injected and flowed over the chip in running buffer (20 mM Hepes, 150 mM NaCl, 0.005% Tween 20, pH 7.4) with or without 0.01 mM $ZnCl_2$. The sensor surface was regenerated with 1.5 M glycine-HCl at the end of each cycle. The data were analyzed using 1:1 binding model with Biacore T100 Evaluation software (GE Healthcare, Chicago, IL, USA).

## DiO labeled viral infection assay

CSFV-infected cell supernatants were centrifuged at 8,000 rpm for 30 min at 4˚C to remove cell debris. Polyethylene glycol (PEG)-8000 (8%, w/v) was slowly added to the supernatants

while stirring at 4˚C for 4 h. The mixture was then centrifuged at 8,000 rpm for 1 h at 4˚C and the pellet was resuspended in PBS overnight. The viruses in the supernatants were purified on 15% sucrose cushion by ultracentrifugation (Beckman SW41 rotor, 35,000 rpm) at 4˚C for 2 h. The pellets were resuspended in PBS and the resulting viral suspension was concentrated using Amicon Ultra-15 100 kDa centrifugal unit (Millipore, Billerica, MA, USA). The virus labeling procedure was performed according to a previous report [49]. Briefly, DiO (Vybrant DiO cell-labeling solution, Invitrogen) at a final concentration of 10 μM and the purified viruses were incubated at room temperature for 10 min while vortexing. The mixture was then buffer-exchanged to PBS to remove unbound dye. PK15 wild-type and *ADAM17*-KO cells were seeded in 24 well plates 24 h before infection. Cells were incubated with DiO labeled virus at MOI of 100 on ice for 1 h and then rapidly warmed to 37˚C for 30 min in order to initiate viral entry. The cells were then fixed with 4% paraformaldehyde and stained with a plasma membrane marker (Wheat germ agglutinin, Alexa Fluor 594 conjugate) and Hoechst 33342. The images were acquired by confocal microscopy (Zeiss LSM 710, Germany). The number of DiO foci in about 100 cells of each cell line were counted.

## Virus binding and internalization assays

The virus binding and internalization procedures were performed according to a previous report [50]. Briefly, for the binding assay, PK15 WT and *ADAM17*-KO Cells ($2 \times 10^5$) were incubated with purified CSFV virions (MOI = 5) on ice for 30 min. After four washes with cold PBS, cells were lysed in TRIzol (Invitrogen, Carlsbad, CA, USA) and RNA was extracted according to the manufacturer's protocol. For the internalization assay, cells were resuspended in fresh media containing 15 mM $NH_4Cl$ and then incubated at 37˚C for 1 h. Following rapid chilling on ice, the cells were treated with 500 ng/ml proteinase K on ice for 1 h to remove excess surface virions. After four washes with cold PBS, cells were lysed in TRIzol and RNA was extracted. The qRT-PCR was performed using a One Step TB Green PrimeScript RT-PCR Kit (TaKaRa, Japan). Primers used for CSFV were: 5'-GAAGGGTAGTCGGCAGGGTCA-3'; 5'-CGGCACCTGTAGCAAGGGTTAT-3'. Primers used for pig β-actin as internal control were: 5'-GCAAGGACCTCTACGCCAACACG-3'; 5'-GGTGGACAGCGAGGCCAGGAT-3'.

## Cell viability assay

Cell viability was determined using the Cell Counting Kit-8 (CCK-8) (GLPBIO, Montclair, CA, USA) according to the manufacturer's protocol.

## Statistical analysis

Statistical analyses were performed using GraphPad Prism software version 8 (GraphPad, La Jolla, CA, USA). The data were expressed as arithmetic means ± standard deviation (SD). The statistical differences between independent groups were assessed by multiple *t* test. *P* values less than 0.05 were regarded as statistically significant.

## Supporting information

**S1 Fig. Targeting sites and sequences of sgRNAs for *ADAM17* and *ADAM10*.** Schematic diagram of *ADAM17* (**A**) and *ADAM10* (**B**) knockout strategy by CRISPR-Cas9. (**C**) Sequences of sgRNAs used for *ADAM17* and *ADAM10*.
(TIF)

**S2 Fig. Overexpression of ADAM17 in KO cells, the effect of flag insertion and biophysical characterization of purified proteins.** (**A**) Over-expression of pig, mouse, human ADAM17

(+pADAM17, +mADAM17, +hADAM17) and hADAM17/H405D mutant in *ADAM17*-KO cells was detected using a polyclonal antibody against hADAM17 by immunofluorescence. Nuclei were stained by DAPI. Size bars indicate 100 μm. (B) Indicated cells were infected with CSFVpp-SXCDK and the titer was measured as FFU/ml. The lower limit of detection was 10 FFU/ml (dashed line). Asterisks indicate samples below the limit. (**C**) Overexpression of flag-tagged pig, chicken and zebrafish ADAM17 (+p*ADAM17*-F, +c*ADAM17* and +zf*ADAM17*) was detected by western blot analysis using anti-Flag antibody. (**D**) Biophysical characterization of sE2 and sMP proteins. Gel filtration profiles of sE2 (left) and sMP (right) proteins were analyzed by size-exclusion chromatography on a Superdex 200 10/300 GL column. The absorbance curves at 280 nm and the SDS-PAGE separation profiles of the pooled samples are shown.
(TIF)

**S3 Fig. Alignment of metalloproteinase domain of ADAM17 from different species by DNAMAN.**
(TIF)

**S4 Fig. Structure of the metalloproteinase domain of hADAM17.** Cartoon structure of the metalloproteinase domain of hADAM17 was prepared from http://www.rcsb.org/3d-view/ 2DDF using PyMOL software. Zinc is colored in purple. Histidine 405 is colored in yellow and blue. Aa301-345 is colored in red.
(TIF)

**S5 Fig. Antiviral effect of aderbasib against CSFV pseudovirus.** (A) Relative pseudovirus infection efficiency in PK15 cells pre-incubated with aderbasib. Briefly, PK15 cells were pre-incubated with various concentrations of aderbasib for 0.5 h, and then infected with CSFVpp or VSV-Gpp in the continued presence of drug. At 48 h after infection, the viral titers were measured as FFU/ml. Results are normalized to infection in the absence of aderbasib. Significance was calculated using a *t* test and *P* values were showed. n.s. = not significant. (B)The relative cell viability was determined using CCK-8. Error bars indicate standard deviation (SD) of the mean (n = 3). The data represent three independent experiments.
(TIF)

## Acknowledgments

We thank Dr. Jianguo Zhao (Institute of Zoology, CAS) for providing the PEF cells, Dr. Enqi Du (College of Veterinary Medicine, Northwest A&F University) for providing the VR1BL cells. We thank Dr. Jianxun Qi (Institute of Microbiology, CAS) for data analysis.

## Author Contributions

**Conceptualization:** Hongkui Deng, George F. Gao, Aihua Zheng.

**Funding acquisition:** Aihua Zheng.

**Investigation:** Fei Yuan, Dandan Li, Changyao Li, Yanan Zhang, Hao Song, Suhua Li.

**Supervision:** Hongkui Deng, George F. Gao, Aihua Zheng.

**Visualization:** Fei Yuan, Dandan Li.

**Writing – original draft:** Fei Yuan, Aihua Zheng.

**Writing – review & editing:** Fei Yuan, Aihua Zheng.

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
