## [Decision Letter · Decision Letter 0]

10 Jan 2021

Dear Dr Zheng,

Your manuscript entitled "ADAM17 is an essential entry factor for classical swine fever virus" has now been seen by three independent referees. As you can see from their comments, both consider that the work is interesting and provides novel insights into the entry process of CSFV. However, in order to further improve the quality of manuscript, some additional issues should be addressed and several additional controls need to be included. Of particular note, reviewer 1 asked to perform specific entry experiments to conclude that ADAM is an authentic CSFV entry receptor.

Sincerely,

Ali Amara

Guest Editor

PLOS Pathogens

Adolfo García-Sastre

Section Editor

PLOS Pathogens

Kasturi Haldar

Editor-in-Chief

PLOS Pathogens

orcid.org/0000-0001-5065-158X

Michael Malim

Editor-in-Chief

PLOS Pathogens

orcid.org/0000-0002-7699-2064

Reviewer Comments (if any, and for reference):

Reviewer's Responses to Questions

**Part I - Summary**

Reviewer #1: In this manuscript Yuan et al. use the CSFV binding protein E2 in a classic biotin-streptavidin pull-down assay to identify putative cellular candidate binding factors. In doing so they identified ADAM17, a cell surface disintegrin and metalloproteinase usually responsible for the cleavage and shedding of growth factors from the cell surface. Deletion of ADAM 17 by CRISPR (exon deletion and frame shift) render the permissive PK15 cell line non-permissive for CSFV binding, which was restored by reintroduction of the ADAM17 gene. They go on to confirm this in multiple cell lines and to show that ADAM17 is not a host determinant. Additional mutation -reintroduction studies show that the metalloprotease domain is important for binding.

While this identification of ADAM17 is a binding factor is interesting, the study does not make any major advance beyond this. In addition, a major issue with this study is that the authors are claiming that ADAM17 is an entry receptor. According to the data, it is not. The data clearly show that it is an attachment factor. Cells lacking ADAM17 show no CSFV binding. One cannot test virus entry when the virus doesn't bind to the cell.

For virus entry it is well established that there are two types of cell surface proteins that have distinct roles: 1-Attachment or binding factors and 2- bona fide entry receptors. Attachment factors serve to bind and concentrate the virus on the cell surface through many high avidity, low affinity interactions and perhaps drive a conformational change in viral fusion proteins. Entry receptors often play no role in virus binding but serve to trigger uptake of the virus through the activation of cellular signaling pathways (hence they are traditionally signaling molecules that span the Pm), they enter the cell with the virus, and they are essential for virus internalization.

Reviewer #2: The manuscript of Dr. Zheng and colleagues describes the discovery of ADAM17 as essential entry factor for classical swine fever virus (CSFV) through a pull-down assay. Classical swine fever is a viral disease of swine, which causes significant economic losses. They robustly show that cells lacking ADAM17 display strongly reduced binding of CSFV and are refractory for infection. They show that despite playing an important role in infection, ADAM17 does not determine the narrow host tropism of CSFV. Finally they pinpoint the domain in ADAM17 that is most important for infection through elegant domain swap experiments. This is manuscript describes well-controlled experiments that identify ADAM17 as a novel receptor for CSFV, which is highly significant. This manuscript will be of high interest to reader interested in viral entry mechanisms in general and in particular to scientist studying the important class of viruses belonging to the Flaviviridae family.

Reviewer #3: In this manuscript, Yuan et al. describe the discovery of ADAM17, belonging to the ADAM protein family of disintegrins and metalloproteases, as an important entry factor for the Classical Swine Fever virus (CSFV), a pestivirus within the Flaviviridae family. In spite of the economical importance of CSFV not much is known about plasma membrane proteins that are involved in virus entry to date.

The authors isolate CSFV E2 binding membrane proteins by pull-down assay using CSFV E2 and cell lysate from a cell line binding high levels of recombinant soluble E2 and identify ADAM17 by mass spectrometry. The important role of ADAM17 was confirmed by the analysis of CSFV entry into knockout cell lines and their counterparts trans-complemented with ADAM17 of different species. Importantly, ADAM17 from mammalian origin but not from distantly related species functionally complemented knockout cells. Subsequently the authors analyzed different porcine cell lines and revealed a correlation between ADAM17 function and susceptibility to CSFV infection. The authors then express a soluble metalloprotease domain of ADAM17 and characterize the interaction with recombinant soluble CSFV E2, demonstrating that Zn2+ ions play an important role for binding. Finally, they exploit the availability of functional and non-functional ADAM17 molecules (from mammalian and zebrafish origin, respectively) to determine the binding region by generating chimeric proteins. Using this approach the authors identify a region of ~45 residues that determines CSFV entry.

The findings of this paper provide novel insights into the entry process of an important animal pathogen, which is poorly understood to date. In this context the presented results will be of great impact for the field of virus research. The experiments were performed carefully and exhaustively demonstrate the key role of ADAM17 in CSFV entry. However, in order to further improve the quality of the paper some additional issues should be addressed and several additional controls added.

**Part II – Major Issues: Key Experiments Required for Acceptance**

Reviewer #1: Throughout the manuscript the authors refer to “entry” for many assays that don't actually measure entry but rather binding or a step downstream of entry: If the authors want to conclude that ADAM17 is an entry factor then the need to actually show it's an entry factor by doing specific entry experiments.

A few examples

Figure 2B: This is not an entry assay, it is a gene expression assay. All it shows is that the pseudovirus did not express GFP. This could have occurred due to:

1-no virus binding

2-no virus uptake

3-abberant virus trafficking

4-Inability of the virus to fuse (or exit) endosome

5-defect in gene expression itself

From Figure 2A one would assume the defect is at the level of binding. Again, indicating that ADAM17 is a binding factor, not an entry receptor.

Figure 3 D. Again, not an entry assay, but an infection assay in this case. Depletion of the inhibitor of ADAM17 gene expression boosting infection does not “confirm ADAM17 as a CSFV entry receptor”

Figure 4 and 5. These represent a nice set of molecular biology experiments that show that the ADAM17 metalloprotease domain is required for CSFV binding, but again no entry assay and the results support ADAM17 as a binding factor.

Reviewer #2: 1. The authors show that aderbasib is inhibiting the interaction of E2 with ADAM17. An intriguing option is to use ADAM17 inhibitors as novel antiviral drugs. Could the authors test if aderbasib or other ADAM17 inhibitors are antiviral by testing their effect on viral replication/entry using the pseudovirus assay?

Reviewer #3: 1. Fig. 2G: The experiment presented in this figure is probably the weakest experiment presented in this study. One fluorescence image does not allow to draw conclusions on the comparison of different cell lines (or the effect of a knockout). The number of DiO foci in at least 100 cells should be counted in both cell lines to quantify the observed effect and the resulting diagram should be appended to the figure. In addition, the authors claim that in the knockout cells “…. most virions remaining on the plasma membrane (Fig 1G) …”(P.7, ls 134-135). This is not obvious at all from the figure and in view of ADAM17 acting as a binding receptor as demonstrated by the authors throughout the entire manuscript one would not expect the virus to bind to the plasma membrane anymore. Please clarify and/or rephrase.

Finally, the figure should be referenced as Fig 2G (P.6, line 133 – “(Fig 1G”))

**Part III – Minor Issues: Editorial and Data Presentation Modifications**

Reviewer #1: (No Response)

Reviewer #2: There are some minor typos so it would be good to carefully proofread the manuscript. For example in line 673: “BHK-21” is mentioned but not shown in the graph; line 247 pestiviruses is misspelled.

Reviewer #3: 1. The paper could benefit from a careful proofreading process and detailed language editing.

2. P. 5, ls. 101-102. The amino acid identity between ADAM17 and ADAM10 is only mentioned in the discussion, but would be helpful for understanding here. Please modify.

3. P.7 ls 148-150. Can it be excluded that the flag tag disrupts ADAM17 function at least for infection. While the rationale for introducing this tag is clear, a control using the porcine ADAM17 carrying the identical tag should be used to demonstrate function of flag-tagged ADAM17.

4. P.9 ls 195-197: In this FACS experiment relatively high inhibitory concentrations are used. Can unspecific effects and/or toxicity be excluded? Appropriate controls and/or a second assay format corroborating the obtained results should be included.

5. P.31 ls 655-656: Should that be “Histidine 405”? Please correct

6. P.32 ls 668-669: “The membrane proteins were labeled with sulfo-NHS-LC-biotin, and blotted with HRP-conjugated streptavidin. Error bars indicate standard deviation (SD) of mean (n=3). The data represent three independent experiments”. Is this referring to panel D? Please clarify and/or rephrase

7. Figure 2: For clarity individual panels in the same figure showing similar experiments or using the same panel of cell lines should follow the same order (here panels A, B and C display different orders – i.e., PK15, ADAM10-KO, ADAM17-KO … in panel A and PK15, ADAM17-KO ADAM10-KO …. in panels B+C).

8. P.34 ls 684-686: The term “CSFVcc infectivity” appears confusing here as the quantified parameter is CSFV RNA. Please clarify in the figure legend, the figure itself and the main text.

PLOS authors have the option to publish the peer review history of their article (what does this mean?). If published, this will include your full peer review and any attached files.

Reviewer #1: No

Reviewer #2: No

Reviewer #3: No
---

## [Editor Report · Decision Letter 1]

15 Feb 2021

Dear Dr Zheng

We are pleased to inform you that your manuscript 'ADAM17 is an essential attchment factor for classical swine fever virus' has been provisionally accepted for publication in PLOS Pathogens.

Best regards,

Ali Amara

Guest Editor

PLOS Pathogens

Adolfo García-Sastre

Section Editor

PLOS Pathogens

Kasturi Haldar

Editor-in-Chief

PLOS Pathogens

orcid.org/0000-0001-5065-158X

Michael Malim

Editor-in-Chief

PLOS Pathogens

orcid.org/0000-0002-7699-2064

Dear Dr Zheng,

Thank you for contributing your revised manuscript entitled "ADAM17 is an essential attachment factor for classical swine fever virus». It is a pleasure to let you know that your manuscript is now accepted for publication in PLOS Pathogens. Congratulations on this interesting work.
---

## [Editor Report · Acceptance letter]

2 Mar 2021

Dear Dr. Zheng,

We are delighted to inform you that your manuscript, "ADAM17 is an essential attachment factor for classical swine fever virus," has been formally accepted for publication in PLOS Pathogens.

Best regards,

Kasturi Haldar

Editor-in-Chief

PLOS Pathogens

orcid.org/0000-0001-5065-158X

Michael Malim

Editor-in-Chief

PLOS Pathogens

orcid.org/0000-0002-7699-2064